# JUST HOW FLEXIBLE ARE NEURAL NETWORKS IN PRACTICE?

## ABSTRACT

It is widely believed that a neural network can fit a training set containing at least as many samples as it has parameters, underpinning notions of *overparameterized* and *underparameterized* models. In practice, however, we only find solutions accessible via our training procedure, including the optimizer and regularizers, limiting flexibility. Moreover, the exact parameterization of the function class, built into an architecture, shapes its loss surface and impacts the minima we find. In this work, we examine the ability of neural networks to fit data in practice. Our findings indicate that: (1) standard optimizers find minima where the model can only fit training sets with significantly fewer samples than it has parameters; (2) convolutional networks are more parameter-efficient than MLPs and ViTs, even on randomly labeled data; (3) whereas stochastic training is thought to have a regularizing effect, SGD actually finds minima that fit more training data than full-batch gradient descent; (4) the difference in capacity to fit correctly labeled and incorrectly labeled samples predicts generalization; (5) ReLU activation functions enable fitting more data despite being designed to avoid vanishing and exploding gradients in deep architectures.

## 1 INTRODUCTION

Neural networks have the ability to model complex and high-dimensional data distributions. Such neural network models are highly flexible and often contain millions or billions of parameters, enabling them to benefit from extremely large-scale training sets.

Common intuition dictates that neural networks can fit at least as many training samples as they have parameters (Zhang et al., 2016; Arpit et al., 2017; Dziugaite & Roy, 2017). This intuition can be most easily understood through linear regression, where a regressor with more coefficients than training samples forms an underdetermined linear system of equations and can therefore precisely fit any function of the training points. For example, consider that for any training set $\{(x_i, y_i)\}_{i=0}^n$ with $n \leq d$, there exist parameters $\{a_i\}$ such that $f(x) = \sum_{j=0}^d a_j x^j$ has $f(x_i) = y_i$ for all $i$ as long as no two training points contain the same input but assigned different labels.

The theory underlying neural networks is significantly more complicated. A variety of approximation theories bound the number of parameters or hidden units required by a neural network architecture to approximate a certain function class on its domain, which is typically infinite (Hornik et al., 1989; Barron, 1992; Mhaskar & Poggio, 2016). On finite domains, namely a training set, *overparameterized* neural networks with many more parameters than training samples can easily fit randomly labeled data, raising questions regarding how such flexible models can still generalize to new unseen test data (Zhang et al., 2016).

In this work, we step back and ask just how flexible neural networks really are in practice. Although neural networks are theoretically capable of universal function approximation (Hornik et al., 1989), we only find optima during training that are accessible via our training procedure, often leading to significantly reduced flexibility as suboptimal local minima exist (Goldblum et al., 2020). In practice, the ability to fit data depends on many factors, including the data itself, model architecture and size, optimizer, and regularizers. We measure the capacity of a model to fit training data, and we examine the practical effects of such factors on the number of training samples a neural network can fit. Our findings are summarized as follows:

- The optimizers typically used for training neural networks often find minima where the model can only fit training sets with far fewer samples than model parameters. This observation calls into question whether we actually find suboptimal local mimima in practice, contrary to conventional wisdom.

- Convolutional architectures (CNNs) are known to generalize better than multi-layer perceptrons on computer vision problems due to their strong inductive bias for spatial relationships and locality, but we find that CNNs are actually more parameter efficient on randomly labeled data as well, indicating that their superior capacity to fit data does not result from superior generalization alone.

- SGD is thought to have a regularizing effect that improves generalization, yet we find that SGD actually enables fitting more training samples than full-batch gradient descent.

- The ability of a neural network to fit many more correctly labeled samples than incorrectly labeled samples predicts generalization.

- ReLU activation functions enable fitting more training samples than sigmoidal activations, even though ReLU nonlinearities were introduced to neural networks to prevent vanishing and exploding gradients in deep neural networks with many layers.

## 2 RELATED WORK

In this section, we review two lines of research related to neural network flexibility and parameter counting, and we explain their relationships to our own work.

**Approximation theory.** A primary area of early deep learning theory focused on upper bounding the number of parameters or neurons required to well-approximate functions in a particular class, for example uniform approximation of continuous functions on a compact domain (Hornik et al., 1989). Such approximation theories typically focus on infinite domains, for instance arbitrary compact sets (Hornik et al., 1989) or a manifold lying in a high dimensional ambient space (Shaham et al., 2018). The resulting upper bounds are often proved constructively, and the constructions may be specific to a particular neural network architecture, often very shallow networks with only a few layers, limiting their generality. In contrast to these works, we focus on the flexibility of neural networks on a finite domain, namely the training set. Moreover, we don't study the existence of parameter vectors that well-approximate functions in theory, as the precise number of parameters required is unknown, instead opting to empirically measure the number of parameters required to fit training data in practice. This methodology allows us to try any architecture or to inspect the influence of optimizers, and it measures quantities that actually impact neural networks in the wild.

**Overparameterized neural networks and generalization.** Early generalization theories predicted that highly constrained models which fit their training data yet fail to fit randomly labeled data (*i.e.* have low Rademacher complexity or VC-dimension) can generalize to new unseen test data (Vapnik, 1991; Bartlett & Mendelson, 2002). However, these theories fail to account for the exceptional generalization behavior of neural networks since they are often highly flexible and overparameterized, leading to vacuous error bounds (Zhang et al., 2016). Recent work on PAC-Bayes generalization theory explains that highly flexible and overparameterized models can generalize well as long as they assign disproportionate prior mass to parameter vectors which fit the training data (Dziugaite & Roy, 2017; Lotfi et al., 2022). Related empirical works explain why neural network inductive bias and consequently generalization can actually benefit from overparametrization (Huang et al., 2019; Chiang et al., 2022). We will see in our own experiments that whereas the Rademacher complexity of neural networks is extremely high, they can fit many more correctly labeled samples than randomly labeled ones in practice, and this gap predicts generalization. Nakkiran et al. (2021) use the data-fitting capacity of neural networks to understand the double-descent phenomenon. In their study, the authors train neural networks with many fewer or many more parameters than the number of samples they can fit, and they study the impact of such over- and underparameterization on generalization. In contrast, we are interested in what influences that capacity to fit data itself.

## 3 PRELIMINARIES

**Quantifying capacity.** We can directly compute the number of samples a linear regression model can fit by counting its parameters. We want to mirror this analysis but for neural networks, measuring the number of samples the model can fit. We require a prospective metric to possess three additional properties: (1) the metric must measure ability to fit data in practice, rather than the theoretical existence of parameters which fit the data, as this property will allow us to compare optimizers and regularizers rather than the hypothesis class alone; (2) the metric must be dataset-dependent so that we can measure the ability to fit data of some types, or with certain labelings, over others; (3) the metric must be easy to compute.

To that end, we estimate the largest sample size where the model can perfectly fit randomly sampled training subsets of that size when optimized until convergence. We call this metric *Empirical Data Capacity* (EDC). We can apply this metric to data sampled in various ways, for example with random or semantic labels or with random inputs.

To compute the EDC, we adopt an iterative approach for each network size. Initially, we start with a small number of samples and train the model. Post-training, we verify if the model has reached 100% training accuracy. If this criterion is met, we re-initialize the model with a **random initialization** and train it again on a larger number of samples, randomly drawn from the full dataset. This process is iteratively performed, increasing the number of samples in each iteration, until the model fails to perfectly fit all the training samples. The largest sample size where the model achieves a perfect fit is taken as the Empirical Data Capacity for that particular network size. It's important to note that each iteration is independent of the others, ensuring that the evaluation of the model's capacity is unbiased and solely dependent on the network size and data subset at hand.

While it is possible to artificially prevent models from fitting their training set by under-training, confounding any study of capacity to fit data, we ensure that all training runs reach a minimum of the loss function by imposing three conditions: first, the norm of the gradients across all samples must fall below a pre-defined threshold; second, the training loss should stabilize; third, we check for the absence of negative eigenvalues in the loss Hessian to confirm that the model has indeed reached a minimum rather than a saddle point. In Appendix Appendix A.3, we detail our method for computing the Empirical Data Capacity as well as how we enforce the above three conditions.

EDC is related to Effective Model Complexity from Nakkiran et al. (2021), but we validate that each model reaches a minimum during optimization. This step is important given that we train models of various architectures and sizes, so we want to prevent under-training from being a confounding variable.

**Underparameterization and overparameterization.** Linear models are described as *underparameterized* when they have fewer parameters than training samples and *overparameterized* when they have more parameters than training samples. This threshold determines when a linear regression model can fit any labeling of its data, and it often coincides with the transition to strict convexity when a linear model has a unique optimal parameter vector. Neural networks behave differently than linear regression models; their loss function is non-convex and can have multiple minima even when training sets are large. Moreover, it is unclear exactly how many parameters a neural network needs to fit its training set in practice. We will use EDC to investigate the latter quantity.

**Capacity, flexibility, expressiveness, and complexity? What's the difference?** These terms are used in numerous ways, sometimes interchangeably and sometimes distinctly. For example, as described above, Rademacher complexity and VC-dimension are notions of complexity which are related to flexibility, whereas the PAC-Bayes notion of complexity instead measures compression. Expressiveness can be used to described the breadth of an entire hypothesis class, that is all the functions that a model can express across all possible parameter settings. Approximation theories measure the expressiveness of a hypothesis class by the existence of elements of this class which well-approximate functions. We will abstain from using the terms "expressiveness" and "complexity" when describing EDC to avoid confusion, and we will use "capacity" and "flexibility" when referring to a model's ability to fit data in practice.

**What can influence EDC?** In contrast to VC-dimension or notions of expressiveness employed by approximation theories, EDC depends not only on the hypothesis class but on every aspect of neural network training, including the optimizer, regularizers, and the exact parameterization of the hy-

pothesis class induced by the model architecture. Architectural design choices shape the geometric properties of the loss surface and make some solutions more accessible than others. Meanwhile, training algorithms influence the types of minima we find during training.

# 4 EXPERIMENTAL SETUP

We conduct a comprehensive dissection of the various factors that contribute to the flexibility of neural networks. To this end, we consider a variety of network architectures, optimizers, and datasets, which we describe in this section.

## 4.1 DATASETS

We conduct experiments on a variety of datasets, including vision datasets - MNIST (Deng, 2012), CIFAR-10, CIFAR-100 (Krizhevsky et al., 2009) and ImageNet (Deng et al., 2009) and tabular datasets - Forest Cover Type (Blackard & Dean, 1999), Adult Income (Becker & Kohavi, 1996) and Credit dataset (Kaggle, 2021). Since these datasets are relatively small and large models can easily fit all training samples, we also use larger synthetic datasets generated through the Efficient Diffusion Training via Min-SNR Weighting Strategy (Hang et al., 2023), which produces diverse ImageNet-quality samples at a $128 \times 128$ spatial resolution. Specifically, we create datasets dubbed ImageNet-20MS containing 20 million samples across 10 classes.

Since data augmentations inhibit the ability of models to fit data to an extent that depends strongly on the exact augmentation strategy, dataset, and model architecture, we train without augmentation.

## 4.2 MODELS

We evaluate the flexibility of diverse architectures including Multi-Layer Perceptrons (MLPs), convolutional networks such as ResNet (He et al., 2016) and EfficientNet (Tan & Le, 2019), and also Vision Transformers (ViTs) (Dosovitskiy et al., 2020). We also explore different scales by systematically increasing the width and depth of these architectures. For MLPs, when we increase only the width of the network, we fix the number of layers and increase the number of neurons in each layer; conversely, when we scale only the depth, we fix the number of neurons in each layer. For naive convolutional networks, we employ several convolutional layers followed by a fully connected layer of constant size. We either change the number of filters in each convolutional layer or the number of layers. For ResNets, we scale the number of filters or the number of blocks (depth). For ViTs, we scale the number of encoder blocks (depth), the dimensionality of patch embeddings, and self-attention (width). Unless otherwise noted, we scale width to control parameter count by default.

## 4.3 OPTIMIZERS

To explore the role of optimization on Empirical Data Capacity, we employ several optimizers including Stochastic Gradient Descent (SGD), Adam (Kingma & Ba, 2015), AdamW (Loshchilov & Hutter, 2018), full-batch Gradient Descent (GD) and the second-order Shampoo optimizer (Anil et al., 2021). These choices enable us to scrutinize how different features such as stochasticity and preconditioning influence the minima we find in practice. In order to facilitate optimization on various dataset and model sizes, we tune the learning rate and batch size carefully for each setup, omitting weight decay in all cases. Our hyperparameter tuning setup is found in Appendix A.2

# 5 THE RELATIONSHIP BETWEEN MODELS AND THE DATA

**Flexibility across data modalities.** We initiate our analysis by measuring the Empirical Data Capacity of neural networks on various training datasets. Specifically, we train a 2-layer MLP on a variety of image classification (MNIST, CIFAR-10, CIFAR-100, ImageNet) and tabular (CoverType, Income, and Credit) datasets. The model is scaled by modifying the width of the hidden layers. Our findings reveal significant disparities in the EDC of networks trained on these different types of data (see Figure 1 (left). For instance, networks trained on the tabular datasets exhibit higher capacity. On image classification datasets, we observe a strong correlation between typical test accuracies

on the task and capacity. Notably, MNIST yields the highest EDC, whereas ImageNet shows the lowest, pointing to the relationship between generalization and the ability to fit data, which we will expand on momentarily.

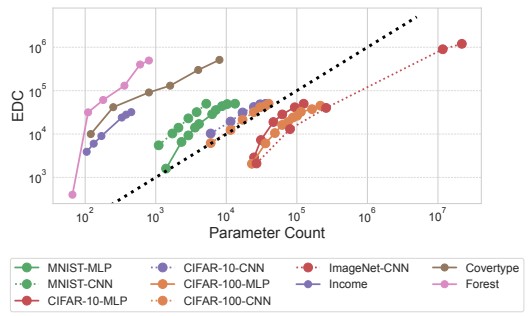 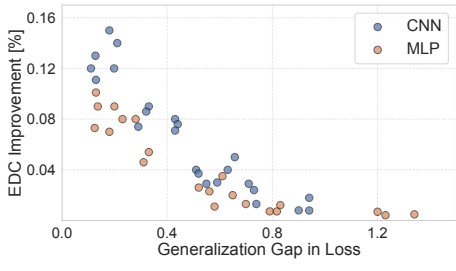

Figure 1: **Left -** EDC compared across datasets and data modalities. Tasks where models generalize well tend to have high EDC. **Right -** The difference in EDC on the original labels and random labels predicts generalization.

**What can random inputs and labels reveal about neural network inductive biases?** We now alter the inputs and their labels to observe their impact on flexibility. We train both MLP and 2-layer convolutional networks with varying widths on the ImageNet-G20M dataset, by adjusting the number of neurons (MLPs) and the number of filters (CNNs) in each layer. We then plot the Empirical Data Capacity as a function of the model's parameter count, considering four scenarios: (1) naturally occurring (semantic) labels, (2) random labels, (3) random inputs, and (4) inputs under a fixed random permutation. In the case of random labels, we maintain the input structure but sample the class labels randomly. For random input, we replace the original inputs with Gaussian noise, while for the permuted input we use the same fixed permutation for all the images.

**The boundary between over- and under-parameterization for neural networks.** Linear regression models can fit at least as many samples as they have parameters, regardless of whether the labels are naturally occurring or random. The boundary between where a model has too few parameters to fit its data and where it has extra degrees of freedom is clear for linear regression. Naturally occurring labels present a more complicated scenario; for instance, if the data's labels are a linear function of the inputs, then the model can fit infinitely many samples. In Figure 2, assigning random labels instead of real ones allows us to explore an analogous notion of the boundary between over- and under-parameterization, but in the context of neural networks. We see here that the networks fit significantly fewer samples when assigned random labels, indicating that neural networks are less parameter efficient than linear models in this setting. Like linear models, the amount of data they can fit appears to scale linearly in their parameter count.

**Ability to fit more semantic labels than random ones predicts generalization.** Figure 3 sheds light on a model's inductive bias, or preference for naturally occurring labelings over random ones. Models often can fit significantly more semantically labeled samples than randomly labeled ones because their generalization abilities effect their capacity to fit data too. In some cases, we observe that generalization allows convolutional networks to fit training sets with more samples than they have parameters.

Common machine learning lore suggests that models with high capacity are more likely to overfit the training data, thus generalizing poorly on new data. This idea underpinned early generalization bounds which are vacuous for neural networks (Vapnik, 1991; Bartlett & Mendelson, 2002). PAC-Bayes generalization theory instead says that a highly flexible model will still generalize as long as its prior assigns disproportionate mass to the true labels compared to random ones (Dziugaite & Roy, 2017). Our empirical analysis relates these two theories for generalization by showing that there is in fact an empirical relationship between capacity and generalization.

We compute the EDC of various different sizes of CNNs and MLPs on the original and randomly labeled data. For each model, we measure the percent increase in EDC on semantic labels compared

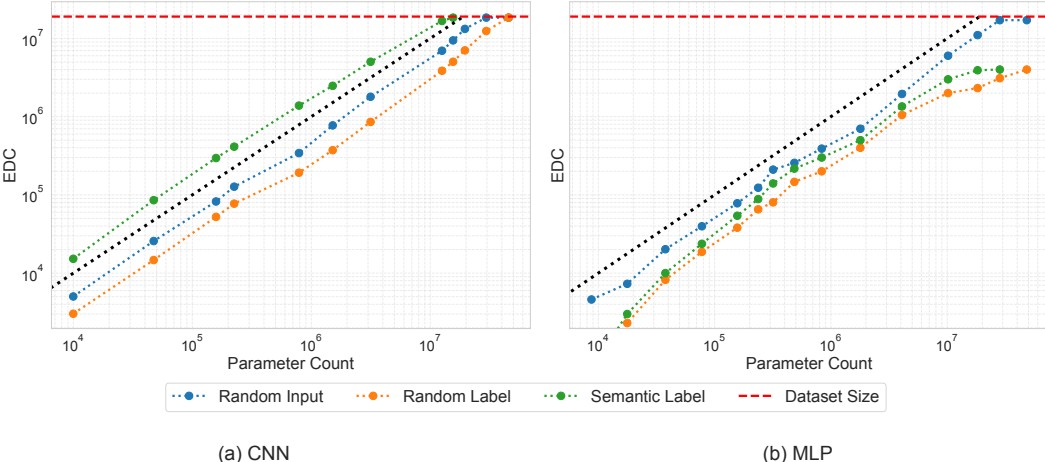

Figure 2: **The influence of generalization on capacity.** Empirical Data Capacity (EDC) as a function of the number of parameters for semantic labels vs. random input and labels for MLPs **(a)** and CNNs **(b)**. ImageNet-20MS dataset. **CNNs fit more semantically labeled samples than they have parameters due to their superior image classification inductive bias, whereas MLPs cannot.**

to random labels. In other words, we measure the extent to which a model has a greater capacity to fit correctly labeled data in practice. In Figure 1 (right), we observe that this quantity has a strong inverse correlation with the generalization gap. We compute a Pearson correlation coefficient of $-0.9281$ for CNNs and $-0.869$ for MLPs. In our experiments, models that can fit many more semantically labeled samples than randomly labeled samples tend to generalize better. While the percentage increase in EDC on semantic labels compared to random labels does not directly measure the additional prior mass assigned to the semantic labels, as PAC-Bayes analysis would demand, we see here that in fact these notions of generalization may be related in practice.

**The distinct inductive biases of MLPs and CNNs.** We see in Figure 2 that both multi-layer perceptrons and convolutional networks can fit significantly more data when assigned semantic labels than when assigned random labels, owing to their generalization ability. The convolutional networks can fit more samples than they have parameters, since they possess a superior inductive bias for image classification compared to the MLPs and hence generalize better.

We glean further differences between the two architectures by manipulating their inputs. CNNs, which benefit strongly from data with a spatial structure, can fit fewer samples when the spatial structure is broken via permutation. On the other hand, MLPs lack this preference for spatial structure, and therefore their ability to fit data is unchanged. When we replace the inputs altogether by Gaussian noise, both architectures exhibit an increased capacity. This trend might be explained by the fact that in high dimensions, noisy data lies very far apart and is therefore easier to separate. Notably, CNNs can fit far more samples with random labels than random inputs whereas we see this trend reversed for MLPs, again highlighting the superior generalization of CNNs on image classification.

**Convolutional networks are more parameter efficient than MLPs and transformers.** There is an ongoing debate regarding the efficiency and generalization capabilities of convolutional networks and Vision Transformers (ViTs) (d'Ascoli et al., 2021; Patro & Agneeswaran, 2023; Maurício et al., 2023). In light of this, we consider three neural network architecture paradigms: (1) MLPs, (2) CNNs, and (3) ViTs. These architectures embody different paradigms in neural network design. We again measure their Empirical Data Capacity across varying model sizes, modifying the width of each architecture and observing the increase in EDC.

In Figure 3a, we see that CNNs, characterized by hard-coded inductive biases in the form of locality and weight sharing, showcase superior EDC on semantically labeled data across all scales in comparison to ViTs followed last by MLPs. One might think that this ordering occurs because CNNs generalize better than ViTs which in turn generalize better than MLPs. To put this hypothesis to the test, we conduct this same experiment on randomly labeled data. We see that in fact

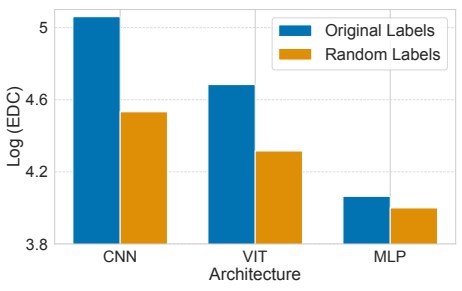 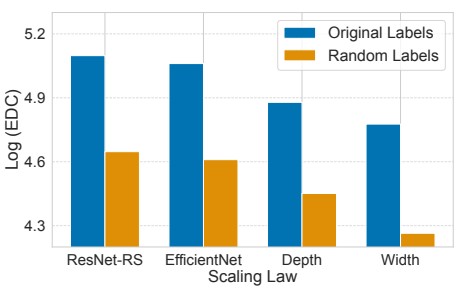

(a) **CNNs have higher EDC**  (b) **EfficientNet is an efficient scaling strategy**

Figure 3: Average logarithm of the Empirical data capacity across parameter counts for different architectures **(a)** and scaling laws **(b)** using original and random labels - ImageNet-20MS dataset. **CNNs are more parameter-efficient than their competitors, even on randomly labeled data.**

this same ordering persists distinctly even on random labels, indicating that generalization cannot explain the superior parameter efficiency of convolutional networks. This result highlights a relationship between EDC and approximation theory, where convolutional networks are thought to be more parameter-efficient than MLPs (Bao et al., 2014).

**Scaling width and depth to optimize capacity.** The question of how to optimize the capacity of neural networks given a parameter count brings to light a classical debate within neural network research — which holds more value, depth or width? While much of the previous work aims to analyze or empirically assess which network exhibits better generalization performance under different scaling laws or explores the theoretical capacity of these networks Bello et al. (2021); Zhai et al. (2022), we unravel a different question — which rule for scaling depth and width enables a network to fit more data in practice? In Figure 3b, we examine the Empirical Data Capacity across various scaling configurations of the networks— either solely augmenting the width (by increasing the number of filters), solely enhancing the depth, or escalating both the width and depth simultaneously using two distinct scaling laws — the EfficientNet scaling law (Tan & Le, 2019) and the ResNet-RS scaling law (Bello et al., 2021). The EfficientNet scaling law proposes a balanced scaling approach, where the depth, width, and resolution of the network are scaled simultaneously in a manner that keeps the total FLOPs nearly constant. Specifically, given a target scaling factor $\alpha$, the depth $d$, width $w$, and resolution $r$ are scaled according to the relations $d = \alpha^{2.0}$, $w = \alpha^{1.0}$, and $r = \alpha^{0.5}$, respectively. For the ResNet-RS scaling law, Bello et al. (2021) proposed a scaling that depends on the training regime, taking into account the model size, number of epochs, and the amount of data, and accordingly changing the image resolution, width, and depth of the network.

Our analysis reveals that, although not initially crafted for optimizing capacity, the EfficientNet scaling law also performs well in this respect. Furthermore, consistent with earlier theoretical analyses (Eldan & Shamir, 2016), our findings affirm that scaling the depth of networks is a more parameter-efficient strategy compared to scaling the width. These parameter-efficiency comparisons also hold on randomly labeled data, indicating that they are not an artifact of generalization. This exploration not only emphasizes the profound impact of scaling strategies on a network's capacity but also adds to the broader understanding of effective network design, assisting in the meticulous endeavor of developing architectures that demonstrate both enhanced capacity and efficient parameter utilization.

**ReLU networks can fit more data.** Nonlinear activation functions are crucial for neural network capacity because without them, neural networks are just large factorized linear models. In this subsection, we delve into the effect of different activation functions on capacity, and we contrast them with deep linear models.

In Appendix Figure 16, we compare the Empirical Data Capacity of CNNs employing ReLU, tanh, and identity activation functions. We see that ReLU activation functions, despite being added to neural networks to avoid vanishing and exploding gradients, exhibit superior capacity due to inherent generalization capabilities. In contrast, we see that randomly labeled data, deep linear models, neural networks without nonlinear activation functions, are significantly more parameter efficient than nonlinear models until their EDC plateaus when their parameter count reaches the number of

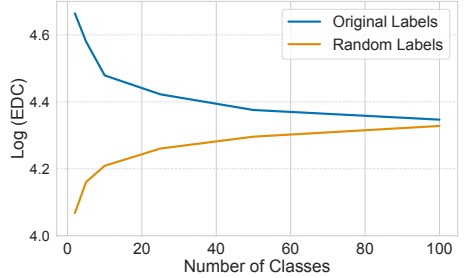 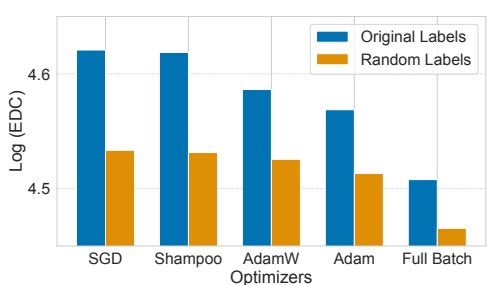

(a) **More classes makes fitting data harder with semantic labels but easier with random ones.**

(b) **SGD and Shampoo are better for fitting with the original labels but not with random ones**

Figure 4: **The effect of the number of labels and of optimizers on capacity.** Average of the logarithm of the Empirical Data Capacity (EDC) across different model sizes for original and random labels varying numbers of classes **(a)** and for different optimizers **(b)**. CNN architectures on the CIFAR-100 dataset.

input features times the number of classes. Since the deep linear model is simply a factorized linear classifier, its capacity has a hard limit regardless of width and depth.

**High-dimensional data is harder to fit with semantic labels but easier with random ones.** Linear models can fit more data when their input contains more features because their parameter count scales with the number of features. In our convolutional neural network setup, we avoid adding parameters as the data dimensionality increases by employing average pooling prior to the classification head, a standard technique for CNNs. We measure the Empirical Data Capacity on ImageNet-20MS by resizing the inputs in order to vary the spatial dimensions from $16 \times 16$ to $256 \times 256$. In contrast to linear models, we find in Appendix Figure 17 that convolutional neural networks, which do not benefit from additional parameters as the input dimensionality increases, can actually fit more semantically labeled data in lower spatial dimensions. This observation also mirrors that of Pope et al. (2020), who find that convolutional networks generalize better on data with low intrinsic dimension.

**More classes make fitting data harder with semantic labels but easier with random ones.** In order to probe the influence of the number of classes on the Empirical Data Capacity, we can randomly merge CIFAR-100 classes to artificially decrease the number of classes while still preserving the size of the original dataset. We again consider a 2-layer convolutional network with various numbers of filters, and consequently, parameters. In Figure 4a, we plot the average of the logarithm of the Empirical Data Capacity (EDC) across different model sizes for various numbers of classes. We see that data with semantic labels becomes harder and harder to fit as the number of classes increases, and generalization becomes more challenging as the model has to encode more information about each sample in its weights. In contrast, randomly labeled data is easier to fit as the number of classes increases because the model is no longer forced to assign as many semantically different samples the same class label, which would be at odds with the model's inductive bias that prefers correct labels over random ones.

## 6 THE ROLE OF OPTIMIZATION IN FITTING DATA

In addition to the architecture and the data we train on, the optimizer and regularizers we choose may have an impact on the types of optima we find during training. We now scrutinize the role of various optimization and regularization techniques in flexibility.

**Stochastic optimizers fit more data.** We begin by comparing SGD, full-batch gradient descent (GD), Adam (Kingma & Ba, 2015), AdamW (Loshchilov & Hutter, 2018), and Shampoo (Gupta et al., 2018) which is a heavily preconditioned optimizer designed specifically for neural networks. Whereas previous works suggest that SGD has a strong flatness-seeking regularization effect (Geiping et al., 2021), we find in Figure 4b that SGD also enables fitting more data than full-batch (non-stochastic) training, fitting a comparable volume of data as the high-powered Shampoo. This ex-

periment, namely the variety of EDC measurements across optimizers, demonstrates that optimizers differ not only in the rate at which they converge but also in the types of minima they find. Repeating this experiment with random labels shows that the higher EDC of SGD and Shampoo evaporates, indicating that their greater ability to fit data may be related to their superior generalization.

**Regularizers - capacity control or better generalization at no capacity cost?** Classical machine learning systems employed regularizers designed to reduce capacity. For example, ridge regression applies a penalty on the parameter norm, improving performance of overparameterized linear models (Hoerl & Kennard, 1970). Similarly XGBoost penalizes the sum of squared leaf weights to prevent overfitting (Chen & Guestrin, 2016).

Modern deep learning pipelines similarly employ a number of regularization techniques that improve their generalization. We now determine if these regularizers designed for deep learning similarly reduce the model's capacity to fit data. We already saw previously that stochastic training, which enhances generalization and is hypothesized to provide implicit regularization, actually increases EDC.

In Appendix Figure 15, we compute the EDC of a convolutional network trained on the ImageNet-20MS dataset using Sharpness-Aware Minimization (SAM) (Foret et al., 2020), weight decay, and label smoothing (Müller et al., 2019). While weight decay and label smoothing do limit capacity, we observe that SAM actually improves generalization without reducing capacity, even on randomly labeled data. Label smoothing actually modifies the loss function and therefore, a model trained with the smoothed objective may not find minima of the original non-smoothed loss. In contrast, SAM does not influence the loss function itself and simply finds different types of minima than SGD, namely ones which generalize better at no capacity cost.

# 7 REPARAMETERIZATION FOR INCREASED PARAMETER EFFICIENCY

In previous sections, we inspected the impacts of various components of the neural network pipeline on parameter efficiency. We saw that neural networks often fit far fewer samples in practice than they have parameters. In this section, we try to close this gap. To close this gap, we adopt a *subspace training* technique (Lotfi et al., 2022), designed for model compression, whereby we take a CNN's parameter vector and randomly project it into a lower-dimensional subspace, training instead in the lower dimensional space. We also try a quantization experiment where we train the CNN in 8-bit precision instead of the standard 32-bit precision, but with 4 times as many parameters. This quantization experiment deviates from our other parameter-count studies. Here, we count a model with $4 \times n$ 8-bit parameters with a model containing $n$ 32-bit parameters as they are specified by the same number of bits.

In Appendix Figure 19, we observe that subspace training significantly improves parameter efficiency on both semantic and random labels, highlighting that neural networks are wasteful of parameters in practice. We similarly find that quantized models are more efficient on a per-bit basis. In Appendix Figure 20, we see that subspace training closes the gap between a model's parameter count and EDC on random labels prior to subspace training. Interestingly, we also see here that an 8-bit quantized model can fit a quarter times as many randomly labeled samples as it has parameters, closing the same gap on a per-bit basis.

# 8 DISCUSSION

Our findings indicate that parameter counting alone is not a useful tool for determining the number of samples a neural network can fit, or the boundary between underparameterization and overparameterization. Instead, many factors contribute to the EDC including virtually all components of a training routine as well as the data itself. Moreover, we must re-evaluate our understanding of why these components work. We saw that architectural components like ReLU activation functions may solve additional problems that they weren't designed for, and stochastic optimization, for example, actually finds minima where we fit more training samples. Finally, the results of Section 7 suggest that neural networks are wasteful of parameters, and new parameterizations may increase their efficiency.

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

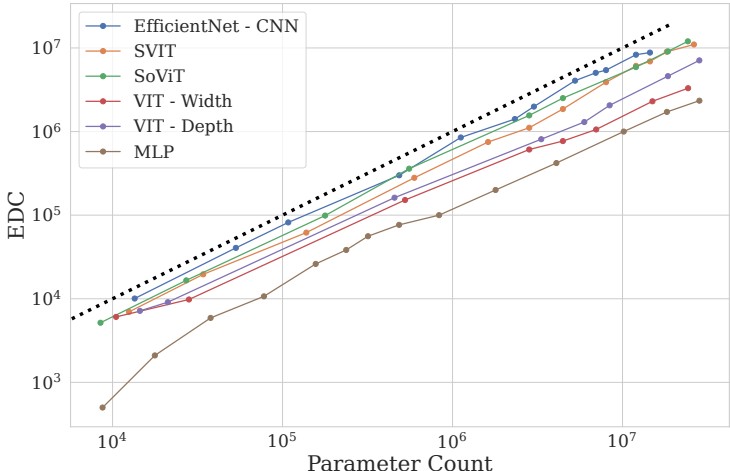

Figure 5: **CNNs have better flexibility on randomly labeled samples -** Empirical Data Capacity as a function of the number of parameters for ViTs, CNNs, and MLPs on ImageNet-20MS dataset with random labels.

# A APPENDIX

## A.1 ADDITIONAL RESULTS

Here, we present figures that include additional datasets and labelings, as well as detailed results across all parameter counts, rather than just the aggregated averages shown in the main body. In the main paper, for the ViT scaling laws, we followed the scaling approach proposed by Zhai et al. (2022) (SVIT), which advocates for simultaneously and uniformly scaling all aspects—depth, width, MLP width, and patch size. Additionally, we employed both SoViT, as per Alabdulmohsin et al. (2023), and approaches where the number of encoder blocks (depth) and the dimensionality of patch embeddings and self-attention (width) in the ViT are scaled separately. fig. 5 in the Appendix demonstrates that scaling each dimension independently can lead to suboptimal results, aligning with our observations from the EfficientNet experiments. Furthermore, it shows that SoViT yields results that are slightly different from those obtained using the laws from Zhai et al. (2022).

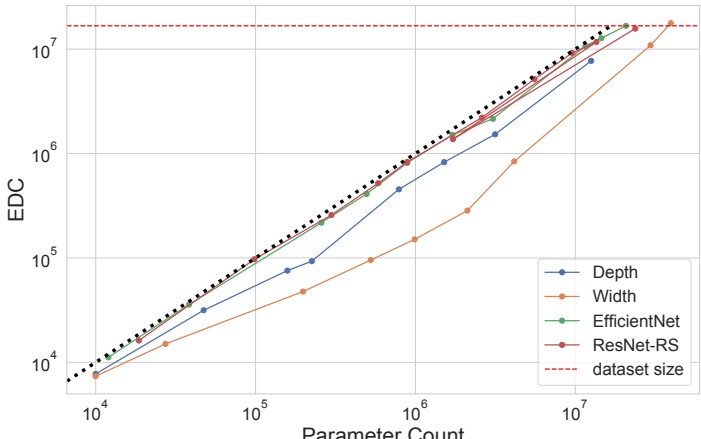

Figure 6: **Scaling laws -** Empirical Data Capacity as a function of the number of parameters for random labels on the ImageNet-20MS dataset.

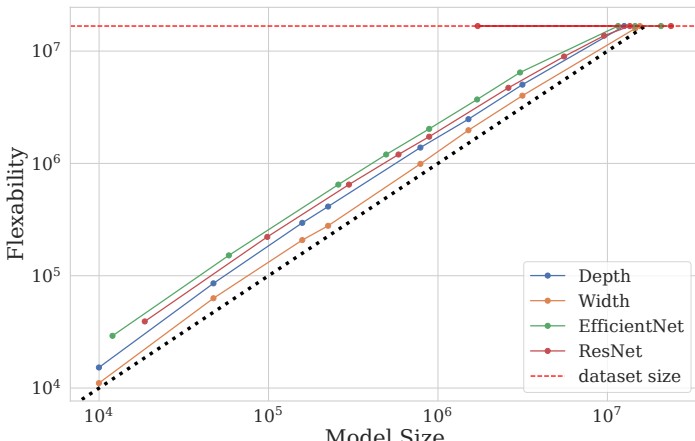

Figure 7: **Scaling laws -** Empirical Data Capacity as a function of the number of parameters for the Original labels with CNN on ImageNet-20MS dataset.

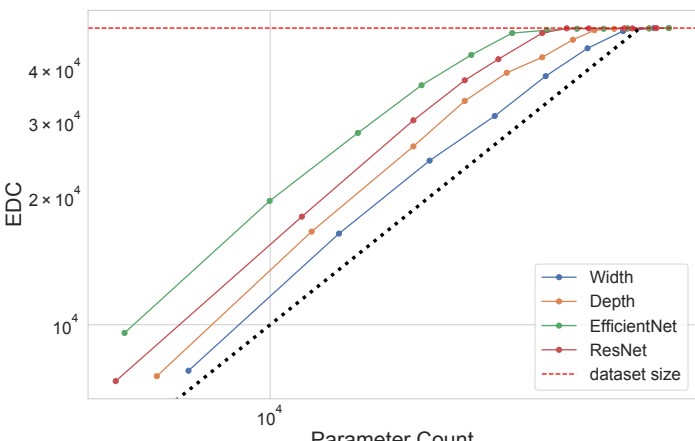

Figure 8: **Scaling laws -** Empirical Data Capacity as a function of the number of parameters for the Original labels with CNN on the CIFAR-10 dataset.

## A.2 IMPLEMENTATION DETAILS

Unless otherwise mentioned, our hyperparameter tuning was conducted over the following hyperparameters: batch size - with the values $[32, 64, 128, 256]$. For the Stochastic Gradient Descent (SGD) optimizer, we used an initial learning rate selected by grid search between $0.001$ and $0.01$ with Cosine annealing. For Adam and AdamW optimizers, the learning rate was chosen by grid search between $1e-5$ and $1e-2$.

For other hyperparameters, we adhere to the standard PyTorch recipes.

## A.3 EMPIRICAL DATA CAPACITY

To compute the EDC (Empirical Data Capacity), we adopt an iterative approach for each network size. Initially, we start with a small number of samples and train the model. Post-training, we verify if the model has perfectly fit all the samples by achieving $100\%$ training accuracy. If this criterion is met, we re-initialize the model with a random initialization and train it again on a larger number of samples, randomly drawn from the full dataset. This process is iteratively performed, increasing the

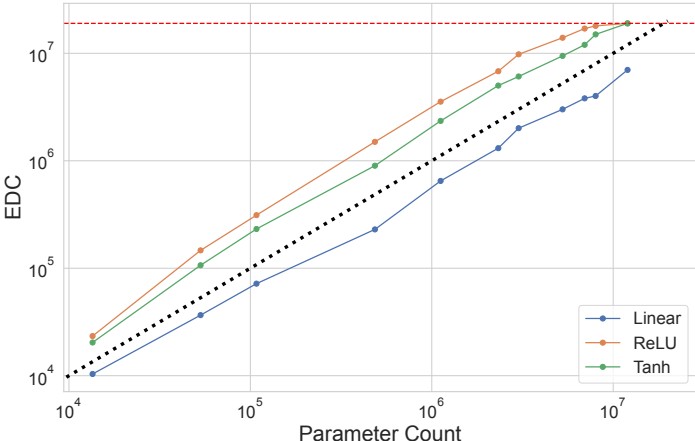

Figure 9: **Empirical Data Capacity as a function of the number of parameters for different activation functions** with CNNs the original labels ImageNet-20MS dataset.

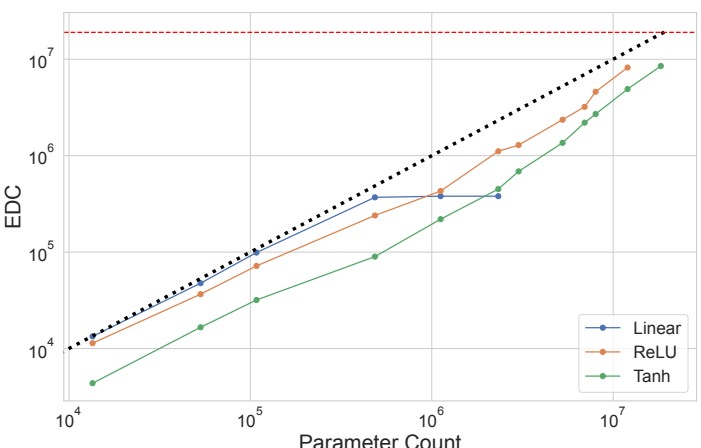

Figure 10: **Empirical Data Capacity as a function of the number of parameters for different activation functions** with CNNs and random labels ImageNet-20MS dataset.

number of samples in each iteration, until the model fails to perfectly fit all the training samples. The largest sample size where the model achieves a perfect fit is taken as the Empirical Data Capacity for that particular network size. It is important to note that data is sampled independently on each iteration.

While it is possible to artificially prevent models from fitting their training set by under-training, thus confounding any study of capacity to fit data, we ensure that all training runs reach a minimum of the loss function by imposing three conditions:

First, the norm of the gradients across all samples must fall below a pre-defined threshold. We observed that there is a high variance in the norms of the gradients between different networks; therefore, we set this threshold manually after checking the norms for each network type when training with a small number of samples, where it's clear that the networks fit perfectly and converge to a minimum.

Second, the training loss should stabilize. To ensure this, we stipulate that the average loss should not decrease for 10 consecutive epochs.

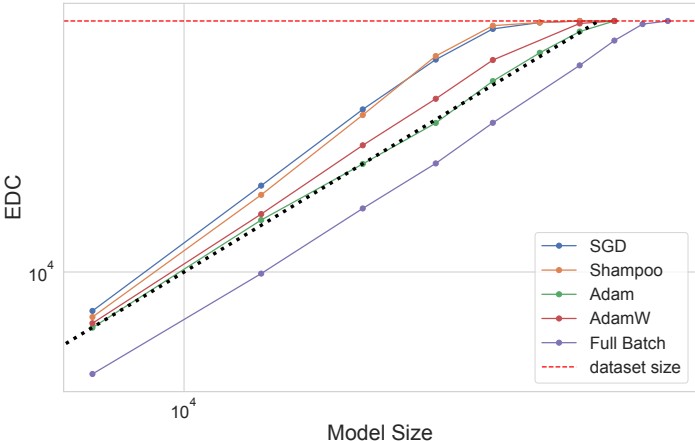

Figure 11: **SGD and Shampoo are better for fitting the training data** - Empirical Data Capacity for different optimizers with CNNs on the CIFAR-10 dataset.

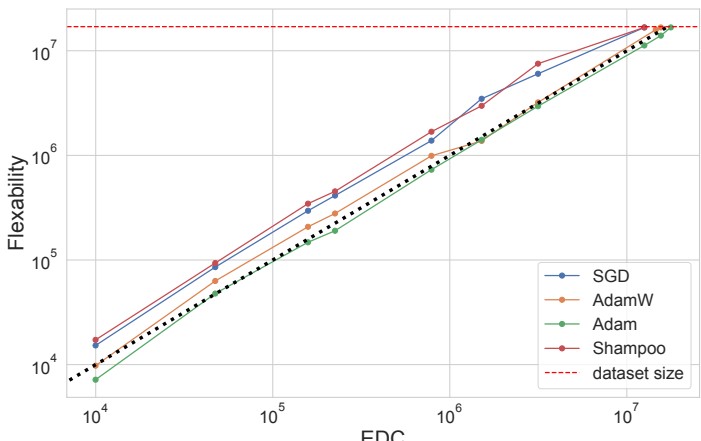

Figure 12: **Empirical Data Capacity as a function of the number of parameters for different optimizers** with CNNs the original labels on the ImageNet-20MS dataset.

Third, we check for the absence of negative eigenvalues in the loss Hessian to confirm that the model has indeed reached a minimum rather than a saddle point. To do this, we calculate the eigenvalues using the PyHessian Python package (Yao et al., 2020) and validate that after training converges, there are no eigenvalues smaller than $-1e - 2$. This threshold was chosen after examining the eigenvalue distributions of different networks that fit perfectly.

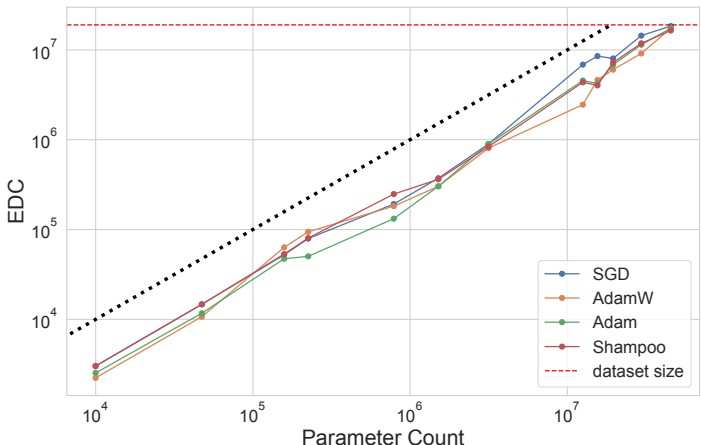

Figure 13: **Empirical Data Capacity as a function of the number of parameters for different optimizers** with random labels on the ImageNet-20MS dataset.

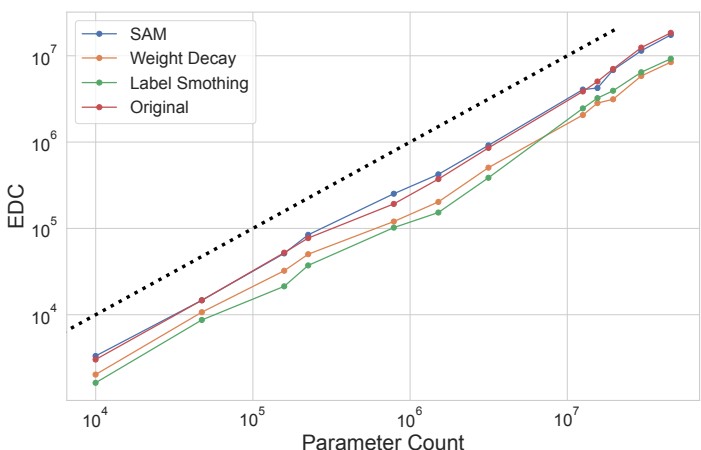

Figure 14: **Empirical Data Capacity as a function of the number of parameters for different regularizers** with random labels on the ImageNet-20MS dataset.

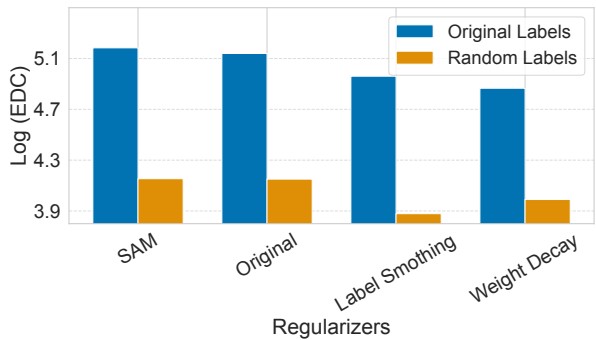

Figure 15: **SAM has better generalization at no capacity cost -** Average logarithm of the Empirical Data Capacity (EDC) over different model sizes for SAM, weight decay, and label smoothing with CNNs on the ImageNet-20MS dataset.

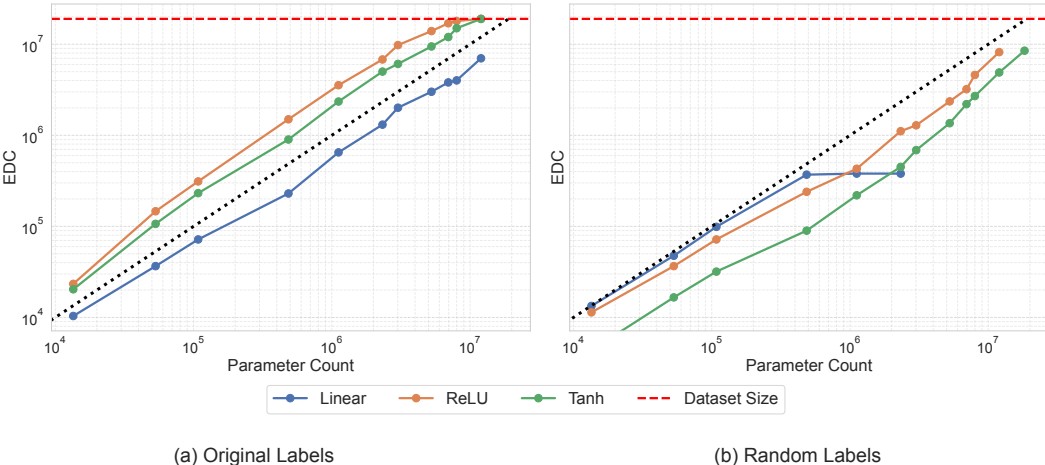

(a) Original Labels          (b) Random Labels

Figure 16: **ReLU networks exhibit higher flexibility.** Empirical Data Capacity as a function of the number of parameters across different activation functions for the original labels **(left)** and for the random ones **(right)** on the ImageNet-20MS dataset.

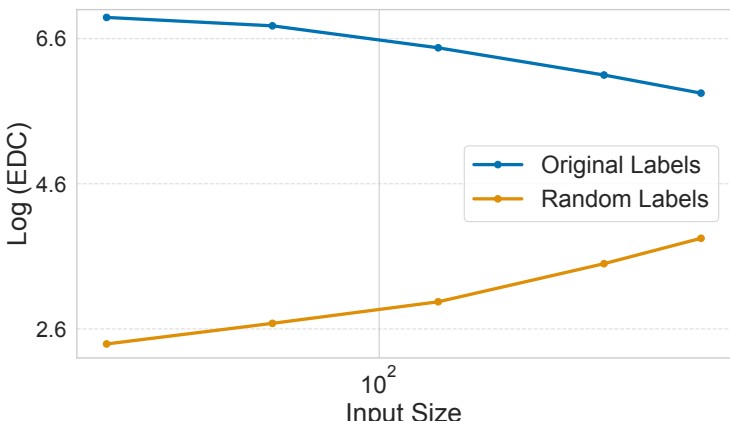

Figure 17: **High-dimensional data is harder to fit.** Average logarithm of the Empirical Data Capacity (EDC) across different model sizes for original and random labels varying input sizes for CNN architectures on the CIFAR-100 dataset.

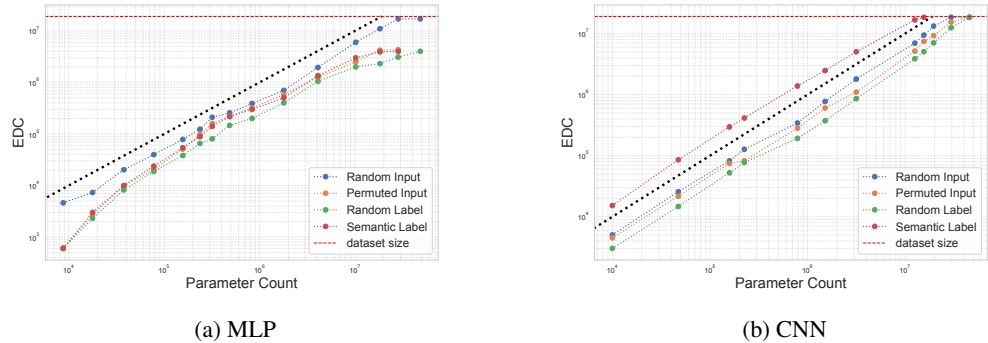

(a) MLP          (b) CNN

Figure 18: **Generalization boosts EDC -** Empirical Data Capacity as a function of the number of parameters for semantic labels vs random input and labels for MLP and CNN architectures on the ImageNet-20MS dataset.

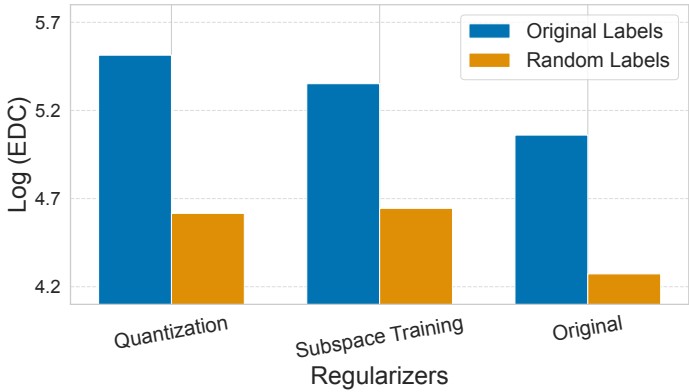

Figure 19: **Compression improves network efficiency -** Average logarithm of the Empirical Data Capacity (EDC) over different model sizes and compression methods. CNNs on the ImageNet-20MS dataset.

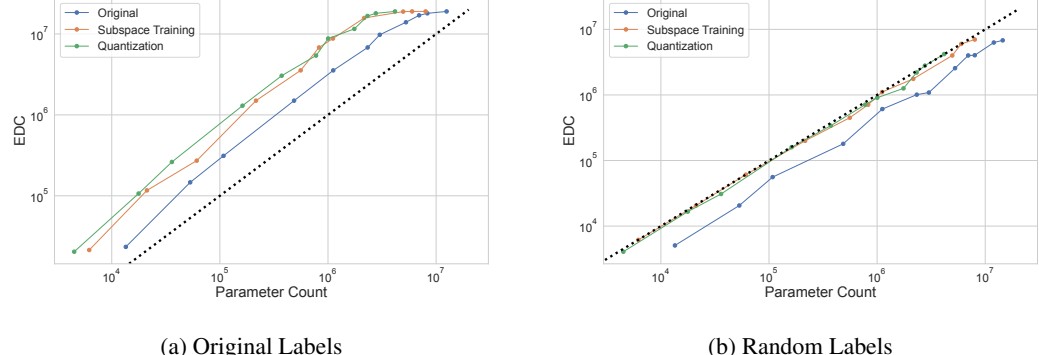

(a) Original Labels          (b) Random Labels

Figure 20: **Compression improves Network efficiency.** The Empirical Data Capacity (EDC) across different model sizes for original and random labels. CNN architectures on the ImageNet-20MS dataset.

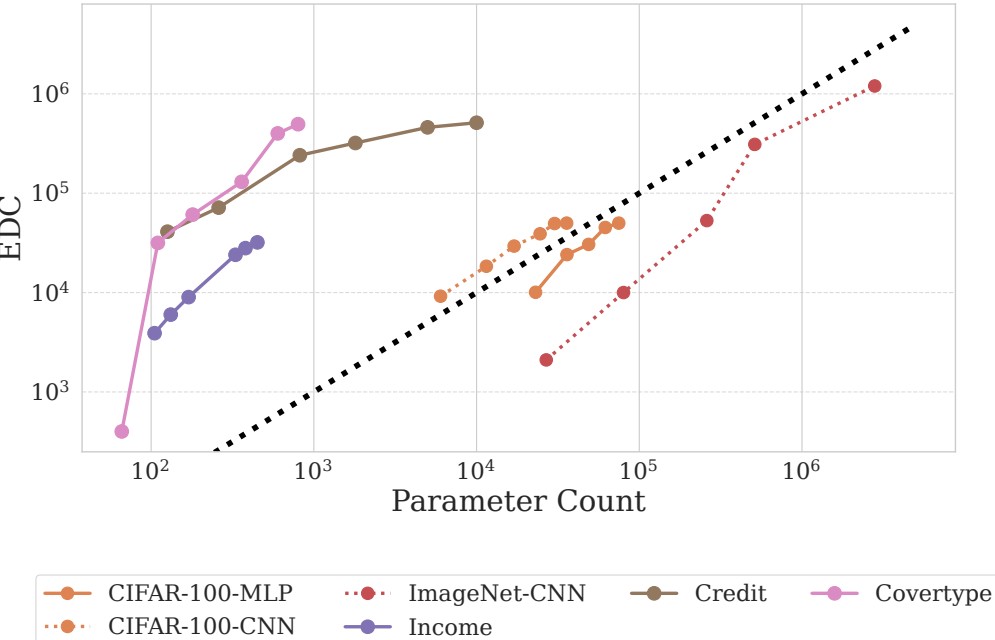

Figure 21: **EDC compared across datasets and data modalities that normalized for binary classification**

