# OpenReview forum: "Just How Flexible are Neural Networks in Practice?"
_ICLR.cc/2024/Conference — Submitted to ICLR 2024_

### Official Review · Reviewer_HFGD · 2023-10-27

**Soundness:** 2 fair
**Presentation:** 3 good
**Contribution:** 3 good
**Rating:** 6
**Confidence:** 4

**Summary:**

This paper studies the flexibility of neural networks, which is defined as the maximum number of training samples that a neural network can accurately classify under sufficient training. The paper analyzes through experiments flexibility as a function of the optimizer, neural network architecture, as well as the dataset. On top of these factors, the paper also considers a large number of variations such as regularization, scaling law, random features and labels, and activation functions. Based on the experimental results, the paper makes several logical conclusions that summarize their findings.

**Strengths:**

1. The topic studied in this paper differs from previous works and has great practical implications. In particular, previous works studying expressiveness solely consider the impact of the number of parameters and the architecture. On the contrary, in this paper, a more fine-grained (and thus more practical) analysis is performed by considering more decisive factors in the training.

2. Some conclusions of the paper are quite interesting. For instance, they found that i). there is a correlation between the increase in EDC going from random labels to semantic labels, and the generalization ability, ii). neural networks are worse than linear models at fitting random labels, and iii). ReLU activations result in higher flexibility though originally designed to mitigate vanishing and exploding gradients, etc.

**Weaknesses:**

1. The measurement of EDC may have some flaws:

    a). The paper takes the strategy of gradually increasing the number of training samples. However, the paper does not discuss how many different trials are run for each setting, and no error bars are plotted. In the very extreme case, there can be some unlucky bad initialization of the neural network that prevented a model from fitting training samples thus causing the EDC to stop growing for that setting. However, this rare case may not be representative of what happens in most cases.

    b). I am not sure why the paper performs the sanity check that each training reached a minimum rather than a saddle point. Since an important consideration of the paper is to evaluate EDC based on the performance of different optimizers, whether the optimizer stopped at a local minimum, global minimum, or saddle points should be a property of the optimizer and it should not be influenced by the experiment design.

2. The paper stops at observation at the surface and does not investigate deeper reasons. For instance, while it is interesting to observe that neural networks are worse than linear models at fitting random labels, the paper should really investigate what is the reason behind such a phenomenon, and potentially answer the question of whether we should really care about this property of NN and linear models in practice. This issue also holds for other conclusions of the paper.

3. Some conclusions of the paper are questionable:

    a). ”Flexibility across modalities”. Is the result that tabular datasets have larger EDC a consequence of simpler tasks?

    b). ”CNN has larger EDC than MLP”. It seems that the experiments are performed only on vision tasks, which makes the conclusion trivial. Does the same result hold on other tasks?

4. Experimental settings are questionable:

    a). I am more sure, when investigating the influence of architecture and over-parameterization, what optimizer settings are the paper using. Is the paper fixing an optimizer, or is the paper trying all optimizers and picking the best-performing one?

    b). Regarding the experiment of scaling up the image sizes, is the filter size also scaled up accordingly?

5. The fact that labels are fit to 100% accuracy may affect the conclusion of some experiments related to generalization since there can be outliers that negatively affect the generalization ability.

**Questions:**

Other than the question I raised in the ”Weaknesses” section, I also want to ask the following: from Figure 1 (right) we see that a larger EDC improvement implies a better generalization ability. Figure 2(b) shows that scaling width under a fixed over-parameterization results in the largest EDC improvement. Do these two results imply that scaling the width leads to the best generalization?

---

> ### Author Response · Authors · 2023-11-21
> **Response to Reviewer HFGD**
>
> Dear Reviewer,
>
> Thank you for your thorough and insightful review of our submission. We appreciate the time you've dedicated to evaluating our work and providing feedback. We address your points below:
>
> 1.  **Measurement of EDD**: \
> **a.  Multiple random seeds:**
> We appreciate the reviewer's input and concur with their observation. To ensure the robustness of our results to random initialization, in the original paper, where our networks did not successfully reach 100% training accuracy, we reran the networks three more times with different random seeds. This was done to ensure that it not fit all the examples. We thank you for highlighting this aspect \
> **b. Consideration of optimization:**
> We agree that we want our study to capture the properties of different optimizers, but in our experiments, none of the optimizers we tried got caught in saddle points.  We chose to check the Hessian spectrum because practitioners train with different optimizers, architectures, datasets, etc, for different numbers of epochs, and we need a way to ensure fair comparison.  Fixing the number of epochs, therefore, would not yield a fair comparison, and simply terminating when the gradient norm is sufficiently low may not yield a fair comparison either because we could end up at a saddle point in the middle of training or a flat region which we would have escaped during the length of a standard training routine, but a gradient-norm based stopping condition would cause us to terminate the training routine very early.
>
> 2. **Deeper Investigation of Observed Phenomena:**
> We struggled to keep the experiments contained in this paper within the allotted space as is. Many of these areas will inspire deeper investigations, and in several cases, those deeper investigations will require their own stand-alone research papers. This paper aims to lay out the broad contours of an area that has been understudied, discovering unexplained phenomena.  We believe that this investigation will contribute significantly to the conference and give the community food for thought.
>
> 3. **About the Conclusions:** \
> **a. Larger EDC for tabular datasets is due to simpler tasks**: There are several ways to define simplicity level, with  pros and cons. However, controlling different factors that may impact the EDC is essential. Inspired by the reviewer's question, we conducted an experiment where we tried to control for properties such as class numbers. We transformed several datasets into binary classification problems, standardizing the number of classes. This modification allowed us to isolate the influence of the input distribution  on EDC more accurately. Our results revealed that the EDC across various image datasets is smaller than tabular datasets (see full details in the the paper). Therefore, we can assert that models can accommodate more examples for each model size for tabular datasets and, by some criteria, can be considered 'simpler'. \
> **b. CNN has a larger EDC than MLP:** CNNs are recognized for their superior generalization on vision problems, attributed to their robust inductive bias for spatial relationships and locality (Taye MM, 2023). However, our work observed that CNNs exhibit greater parameter efficiency even on **randomly labeled data**, suggesting that their enhanced capacity extends beyond better generalization alone.
> Regarding tabular data, in response to the reviewer's inquiry, we investigated the  EDC of MLP and CNNs for tabular datasets. Our findings, detailed in the updated version of our paper, reveal that the EDC is similar for the MLP and CNN. This outcome is unsurprising, as tabular datasets lack the spatial relationships and locality found in computer vision problems. Consequently, we did not anticipate a CNN advantage in fitting more examples than MLP in this context.
>
> 4. **Experimental Settings and Methodologies:** \
> **a. Clarify the optimizer settings**: For our experiments, unless otherwise specified, we utilized SGD for training. We highlighted this detail in the updated version. \
> **b. Scaling filter sizes:** Yes, we increased the filter size proportionally with the input sizes.
>
> 5. **Fitting labels to 100% influenced the experiments:** In principle, it's true, and there might be outliers. During our experiments, we also checked our analysis without requiring 100% accuracy, allowing for a bit less (in order to accommodate outliers), which didn't significantly change our results.
>
> **Questions:** \
> **Regarding the Figures** - Yes, the results imply that scaling the width leads to the best generalization.
>
> Thank you again for your thoughtful review. We made a significant effort to address your feedback, and we would appreciate you raising your score in light of our response.  Please let us know if you have additional questions we can address.
>
>
> ## References
> - Taye MM. Theoretical Understanding of Convolutional Neural Network: Concepts, Architectures, Applications, Future Directions. Computation. 2023

---

### Official Review · Reviewer_14iz · 2023-10-30

**Soundness:** 3 good
**Presentation:** 2 fair
**Contribution:** 1 poor
**Rating:** 3
**Confidence:** 4

**Summary:**

This paper investigates the practical data-fitting capabilities of neural networks, challenging the widely held belief that a neural network can fit a training set as long as it has a sufficient number of parameters. The authors find that the training procedure, including the optimizer and regularizers, as well as the network's specific parameterization and architecture, significantly influence the model's ability to fit data.

**Strengths:**

I think the paper attempts to answer an interesting direction of understanding what makes Neural Networks work in practice an interesting direction of research.

**Weaknesses:**

The paper could benefit from deeper theoretical exploration. While the metric's motivation isn't thoroughly convincing, there's an intriguing mention of PAC-Bayes and VC dimension. It would be valuable to elaborate on how their selected measure aligns with the current theory. While I appreciate the importance of experiment-driven papers, it appears that the conclusions were drawn post-results, rather than being based on predefined hypotheses derived from established theory. The relevance of model architectures to random labels remains unclear, and the frequent references to the linear model are perplexing, especially given that large parameterized neural networks don't typically align with traditional models—consider the double descent phenomenon and their tendency to yield smoother functions [1]. Additionally, section 5's presentation could be improved; its current format reads like a data dump rather than a cohesive exploration of the central theme of 'flexibility'.


[1] - Bubeck, Sébastien, and Mark Sellke. "A universal law of robustness via isoperimetry." Advances in Neural Information Processing Systems 34 (2021): 28811-28822.

**Questions:**

-	The description of the EDC and how it relates to quantifying capacity seems a little bit loose I am not sure why it is a good metric for capacity. Furthermore, it would be good if you could split out step by step the computation.
-	Could you clearly define what you mean by flexibility or an intuitive understanding I found it hard to find a clear definition of what you mean.

---

> ### Author Response · Authors · 2023-11-21
> **Response to Reviewer 14iz**
>
> Dear Reviewer 14iz,
>
> We appreciate your comments and thoughtful considerations regarding the theoretical aspects of our paper. We acknowledge the importance of theoretical exploration in the realm of deep learning; however, we intentionally adopted a different approach in this work.
>
> Our primary motivation was to explore the empirical side of network capacity rather than relying solely on theoretical frameworks such as PAC-Bayes and VC dimension. Historically, attempts to assess the capacity of deep networks have often been theoretical and, at times, not entirely aligned with empirical observations. We aimed to bridge this gap by providing an empirical perspective on network capacity.
>
> While we recognize the value of predefined hypotheses derived from established theory, our focus on drawing conclusions post-results stems from our commitment to an experiment-driven methodology. This approach enhances transparency and clarity in the progress of scientific inquiry. The "double descent phenomenon" (Preetum, et al. 2021)  is a good example; its impact on the community was initially observed empirically, and the (partial) theoretical understanding followed in subsequent works.
>
> Regarding **the relevance of model architectures to random labels and frequent references to linear models**, as mentioned in our paper, many theoretical works provide vague and impractical approximation theory bounds, and this is why it is so important to examine it in practice. We agree that modern neural networks do not align with traditional models, and our paper precisely demonstrates that the practical flexibility of modern networks is much higher than linear models. Concerning random labels, based on previous works (Zhang et al., 2016), we sought to distinguish between the input distribution and the label distribution, exploring how the network can fit the data regardless of the label’s distribution. Moreover, as demonstrated in our paper, the ability of a neural network to fit many more correctly labeled samples than incorrectly labeled samples predicts generalization.
>
> Concerning **Section 5**, we acknowledge your feedback on its current format. We will work on improving the presentation to ensure a more cohesive exploration.
>
>
>
> ### Questions:
>
>
> **EDC Metric Explanation:**
>
> Defining the capacity of a model encompasses various perspectives. Intuitively, it refers to model complexity—a measure of how intricate a pattern or relationship a model can express. In simpler terms, it quantifies the number of examples a model can learn. While VC dimension is one method to define capacity, it may exhibit a considerable gap from the model's actual ability to fit the data and, therefore, not used in practice. In our work, we introduce EDC as an alternative measure for empirical capacity.
>
> In essence, EDC is the maximum number of training data points a model can fit, representing its empirical capacity—the number of patterns it can learn. To elaborate, EDC is defined as the largest sample size where the model can perfectly fit randomly sampled training subsets of that size when optimized until convergence.
>
> As explained in our paper, to compute the EDC, we adopt an iterative approach for each network size. Initially, we start with a small number of samples and train the model. Post-training, we verify if the model has reached 100% training accuracy. If met, we re-initialize the model with random initialization and train it again on a larger number of samples randomly drawn from the entire dataset. This process iteratively increases the number of samples in each iteration until the model fails to fit all the training samples perfectly. The largest sample size where the model achieves a perfect fit is taken as the EDC for that particular network size.  Please refer to section 3 in our paper for a detailed explanation.
>
> **Flexibility:**
>
> We define flexibility as the model's ability to fit the data in practice.  Please refer to the paragraph titled "Capacity, flexibility, expressiveness, and complexity? What’s the difference?" on page 3 for a paraphrased explanation.
>
> Thank you for your valuable insights, and we are committed to refining the paper to address these concerns while maintaining the empirical focus that distinguishes our work.
> We would greatly appreciate it if you raised your score in light of our response.  Please let us know if you have additional questions we can address.
>
> ### References:
> Nakkiran, Preetum, et al. "Deep double descent: Where bigger models and more data hurt." Journal of Statistical Mechanics: Theory and Experiment 2021.12 (2021)

---

> > ### Comment · Reviewer_14iz · 2023-11-22
> > **Response to rebuttal**
> >
> > I think the authors for taking time to address some of my concerns, I however do not feel it fundamentally changes my opinion so will keep my score.

---

### Official Review · Reviewer_ftNw · 2023-10-31

**Soundness:** 3 good
**Presentation:** 1 poor
**Contribution:** 3 good
**Rating:** 6
**Confidence:** 4

**Summary:**

The paper introduces the notion of Empirical Data Capacity (EDC) and study empirically how it behaves across different architectures, optimizers, and data distributions, among others. Informally, EDC is the largest subset of training data that can be correctly classified/interpolated by the neural network when trained without augmentation. Unlike the complexity of the hypothesis space, such as the VC dimension, EDC factors into account the entire training protocol. Borrowing intuition from linear classifiers, it extends the notion of parameter count.

The authors present many interesting findings and insightful discussions. For instance, the authors find that the difference between EDC on true labels and EDC on random labels to be a strong predictor of generalization ($<-0.8$ Pearson correlation coefficient). They also find that the scaling recipe in EfficientNet also improves EDC. Also, more classes make fitting data harder with semantic labels but easier with random ones, and so on. Overall I think it's a good empirical study but would benefit a lot from a more precise discussion/explanation in several places.

**Strengths:**

- The paper is well-written overall and easy to follow. The breadth of the analysis is commendable. The paper additionally presents many interesting results.

**Weaknesses:**

- First, the notion of EDC is identical to the Effective Model Complexity (Nakkiran, et al. 2021), which the authors cite but do not really compare with. Both correspond to the largest subset of training data that is interpolated by the model. However, the authors exercise caution when using it (e.g. by checking the norm of the gradients and the flatness of the loss curve, etc) to make sure that the models are not undertrained. I would recommend that the authors refrain from claiming novelty in the notion of EDC and simply center their contribution around the empirical investigation of the Effective Model Complexity, instead.

- The second primary weakness in my opinion is the lack of clarity about the experimental details in a few places.
  * When comparing different architectures such as CNNs and ViTs, how did the authors ensure that the architectures are compared fairly? For example, there is no mention of the number of parameters used in each architecture. Even if those architectures share the same parameter count, the shape of the architecture can have a big impact as the authors themselves study for CNNs. It's not clear if one can conclude, for example, that CNNs are more parameter efficient than ViTs. There aren't enough details in the paper for the reader.
  * The authors mention repeatedly that ReLU improves EDC even though they "were introduced to neural networks to prevent vanishing and exploding gradients." I do not see the discrepancy here. ReLU were introduced to prevent training instability issues so I would expect ReLU to improve training and improve EDC as a result. It is unclear why the authors highlight that the impact of ReLU on EDC is surprising.
  * When the authors compare different datasets, like CIFAR10/100, MNIST, and ImageNet, they use these results to argue for the impact of the input distribution. However, those datasets have different number of classes. One way to fix that issue is to fix the number of random labels in all datasets. There is no discussion of this in the paper and it seems that the authors kept the label space unchanged. The problem is that it is not clear (again) when the input distribution matters much when accounting for the differences in the number of classes.
  * In many figures, the authors average log(EDC). Is it reasonable to average them? Wouldn't the average EDC be dominated by what happens to large models? Perhaps it would be better to report the raw results without averaging.

- Some minor comments:
  * The author should provide precise definitions of some of the terms they use. For example, what is "parameter-efficient"? My understanding is that it is EDC divided by the parameter count. Is this the case?
  * The figures are not presented in the same order they appear in the main text.
  * Figure 2(b) is very difficult to understand. Are the authors here scaling a single dimension at a time using EfficientNet scaling recipe and reporting the average results?
  * Page 6: typo in "we three neural network"
  * It would be interesting to see of the scaling laws derived for ViT [1] would also increase EDC, similar to the analysis that was done on EfficientNet.

[1] Alabdulmohsin, Ibrahim, et al. "Getting ViT in Shape: Scaling Laws for Compute-Optimal Model Design." NeurIPS, 2023.

**Questions:**

- Can you please clarify how the comparison across different architectures was done? Did you ensure that all architectures have the same size? How were the shapes selected? Are they based on some standard shapes, such as those used in the original ViT and EfficientNet papers? What about MLP?
- When using random labels, did you fix the label space in all datasets (e.g. binary classification) or was the same label space used in each dataset (e.g. 10 classes in CIFAR10 but 100 classes in CIFAR100)?
- Can you please clarify Figure 2(b)?

---

> ### Author Response · Authors · 2023-11-21
> **Response to Reviewer ftNw**
>
> Dear Reviewer ftNw,
>
> Thank you for your detailed evaluation of our submission. We appreciate your comments and will address your concerns and questions below:
>
> 1. **Comparison with Effective Model Complexity (EMC):**
> We appreciate the reviewer's insight on the conceptual resemblance between our Empirical Data Capacity (EDC) and the Effective Model Complexity (EMC) as explored by Nakkiran et al. (2021) and agree that we should focus on our unique contribution to the empirical investigation of the EDC:  Nakkiran et al.'s work defines EMC primarily to dissect the double-descent phenomenon, with a focus on the training procedure and the train/test error of the data. Their exploration is deeply intertwined with the dynamics of under-parameterized and over-parameterized regimes, particularly in relation to test and train error. Conversely, our investigation of EDC takes a distinctive route. We concentrate on the empirical study of neural networks' capacity to fit training data, independent of generalization or double-descent implications. Our methodology for computing EDC includes rigorous measures to assure independence in each evaluation iteration and to prevent under-training. These measures include monitoring gradient norms, stabilizing training loss, and examining the loss Hessian.
> These methodological distinctions underscore that EDC, while sharing the same concept as  EMC, diverges in application, implications, and the questions that it answers. Our focus even extends beyond merely determining the largest subset of data a model can interpolate; it delves into the empirical factors influencing this capacity under varying conditions like architecture and optimizer choices.
> In light of this discussion, we agree with the reviewer's suggestion and will incorporate a detailed comparison between EDC and EMC in the revised version of our manuscript. This addition will further clarify the unique aspects of our study and its contribution to the field.
>
> 2. **Experimental Details and Fair Comparison of Architectures:**
> We apologize for any confusion caused by comparing the average EDC of different architectures in the paper. To provide clarification, for all figures where we plotted the average logarithmic EDC, we have included the complete, unaveraged results in the appendix. Specifically, Figure 5 in the appendix offers a detailed presentation of the EDC across the full spectrum of parameter counts for various models, including VITs, CNNs, and MLP.
> Our decision to feature averaged results in the main text was driven by the consistency of the observed phenomena across different model sizes, where no significant differences were noted between smaller and larger models. This choice aimed to balance clarity and conciseness in the paper's main body while providing extensive data in the appendix for readers seeking more detailed insights.
> Regarding the reviewer’s concern about the impact of scaling laws on the comparison between different models, In our study, we scaled ViTs following the scaling laws proposed by Zhai et al., 2022. These laws advocate for scaling all aspects—depth, width, MLP width, and patch size—simultaneously and uniformly. We have expanded our revised results to include different scaling laws for VIT to address the reviewer's concerns and further elaborate on this aspect. Based on the reviewer's suggestion, we used both SoViT from  Ibrahim et al. (2023) and laws where the number of encoder blocks (depth) and the dimensionality of patch embeddings and self-attention (width) in the ViT are scaled separately.  Figure 22 in the appendix of our updated paper demonstrates that scaling each dimension independently leads to suboptimal results in line with our observations from the EfficientNet experiments and that SoViT has slightly better results to the ones from Zhai et al., 2022.
> This expanded analysis strengthens our original findings and offers a more nuanced perspective on how different scaling strategies influence the EDC of various architectures. This elaboration adequately addresses the reviewer's concerns and provides a more comprehensive understanding of our comparative analysis.
>
>
>
> 3. **Impact of ReLU on EDC:**
> Thank you for pointing out this concern, and we would share your concern that exploding and vanishing gradient problems could in fact cause lower EDC if we were not successfully minimizing the loss function.  Whereas ReLUs were adopted to prevent exploding or vanishing gradient problems that can cause poor conditioning and hamper optimization, we actually ensure that we obtain a minimum of the loss function in each experiment.  That is, we find that ReLUs boost EDC even when we would be able to optimize successfully and converge to minima using competing activation functions.
>
> =>

---

> > ### Author Response · Authors · 2023-11-21
> > **Response to Reviewer ftNw - Part 2**
> >
> > 4. **Dataset Comparisons Considering Class Numbers:**
> > We appreciate the reviewer's observation. Our initial approach was to present the EDC for a variety of datasets and modalities, after which we delved into decomposing the effects of different dataset features on EDC. A key aspect of our exploration was assessing how the number of classes within a specific dataset influences EDC. We found that increasing the number of classes typically makes data fitting more challenging with semantic labels but easier with random labels. This counterintuitive finding underscores the complex interplay between class numbers and data-fitting capabilities. In response to the reviewer's point about the value of comparing datasets while controlling for properties like class numbers, we performed additional analyses for a more nuanced understanding. We converted several datasets into binary classification problems, standardizing the number of classes. This modification enabled us to assess the impact of class numbers on EDC more accurately and isolate the effects of input distribution.
> > Our results from this controlled comparison (Figure 22 in the appendix of the updated paper), showed that even though the EDC among various image datasets increased in the binary classification setting compared to the original setting, tabular datasets consistently demonstrated higher EDC. Furthermore, significant differences persisted among the different tabular datasets. These outcomes suggest that additional factors, perhaps intrinsic to the datasets themselves, significantly contribute to EDC beyond just the number of classes.
> > Incorporating the reviewer's feedback, we will clearly articulate these additional findings and analyses in the revised version of our paper. This enhancement will deepen the understanding of how diverse datasets, each with unique characteristics and complexities, influence the EDC of neural networks.
> >
> >
> > 5. **Methodology for Averaging log(EDC):**  See our comment above.
> >
> > 6. **Precise Definitions:**  We acknowledge the need for precise definitions. Concerning "parameter-efficient," we clarify that a network is considered more parameter-efficient if it consistently demonstrates higher EDC compared to other models across a range of parameter counts.
> >
> > 7. **Figures Order:** We appreciate your attention to detail regarding figure order. The figures have been reordered to align with their appearance in the main text.
> >
> > 8. **Figure 2(b) Clarification:**
> > We acknowledge the difficulty in understanding Figure 2(b). Figure 2(b) illustrates the average results of EDC across different network sizes and not the average across different dimensions of EfficientNet scaling laws. The full, unaveraged data is presented in Figure 6 in the appendix. As discussed in our paper, there are different ways to scale network parameters. This figure presents the EDC for four different scaling laws: width, depth EfficientNet scaling law, and ResNet-Rs scaling law. For the EfficientNet law, we applied the compound scaling method from the EfficientNet paper, adjusting width, depth, and resolution uniformly with a compound coefficient simultaneously. This is in contrast to scaling only the width or the depth, which is also presented in the figure.
> >
> > 9. **Typos:** We thank you for catching the typo. The error has been corrected in the updated version of the paper.
> >
> > 10. **Scaling Laws Analysis:**
> > Based on your comment, we conducted additional analyses following the scaling laws presented by Ibrahim et al. (2023). Similar to the EfficientNet analysis, the scaling laws improved ViT results. However, ViT still underperformed compared to CovNet. For further details, please see our response to Question 4 above.
> >
> >
> > ### Questions:
> >
> > - **Comparison across different architectures:** See our response to Question 2. We calculated their EDC across the entire range of model sizes for all architectures. See Figure 5 in the appendix for detailed results.
> >
> > - **Fixing the label space when using random labels:** In our original results, we did not fix the number of classes. Following your suggestion, we conducted additional experiments by converting all datasets to binary classification. For further details, please see our response to Question 4 above.
> >
> > - **Figure 2(b) Clarification:** See our response in Answer 8 for clarification on Figure 2(b).
> >
> >
> > Thank you again for your thoughtful review. We made an effort to address your feedback, including multiple experiments and paper edits, and we would greatly appreciate it if you would consider raising your score in light of our response.  Please let us know if you have additional questions we can address.

---

> > > ### Comment · Reviewer_ftNw · 2023-11-21
> > > **Thanks**
> > >
> > > Thanks for the detailed reply and for the additional experiments.
> > >
> > > Regarding the connection to EMC, your response does not show that EMC is a different measure from EDC. Instead, it is the same notion, except that you take extra care in calculating it and use to study different problems. This is fine and doesn't hurt the contribution of the work, but giving it a different name is misleading and (in my opinion) unfair to the authors of EMC.
> > >
> > > One minor comment about Figure 1: the distinction between MLP and CNN is hard to see. It may be better to use light colors and dark colors.
> > >
> > > Overall, I'm raising my score given that the paper contains findings that are quite interesting, and that most of my concerns have been addressed in the revision. I still have a strong reservation against renaming EDC with EMC.

---

### Meta-Review · Area_Chair_PEjw · 2023-12-10

**Metareview:**

This paper uses a quantity known as "empirical data capacity" (EDC) to study the test error of deep networks using a number of architectures, datasets and optimizers. The EDC is the largest number of training samples that can be fit perfectly by a network and optimizer. EDC is similar to effective model capacity of Nakkiran et al. 2021. The value of such empirical papers is well appreciated. But it is difficult to distill the conclusions from the present manuscript. Many of the claims that are made by the authors are either too vague or are not substantiated with clear evidence which disentangles the effect of the numerous moving parts in the problem, e.g., typical optimizers find local minima where many samples in the dataset are not fitted, SGD fits more samples than gradient descent, generalization is correlated with the difference between performance on "typical" data and random data, ReLU nonlinearities enable fitting more samples than sigmoidal nonlinearities. As another example, the paper claims that tasks where models generalize have high EDC; this is not true as written. The reviewers have also raised a number of these concerns.

As such, the manuscript has interesting experiments but it falls short of drawing precise conclusions from these experiments. It would also be useful to connect these findings to existing theoretical analyses of generalization.

**Justification For Why Not Higher Score:**

This manuscript might be the beginning of a good paper but, as of now, it well below the bar for intellectual novelty, clarity and scientific contribution. The claims in the paper should be precisely stated and substantiated with a thorough analysis.

**Justification For Why Not Lower Score:**

N/A

---

### Decision · Program_Chairs · 2024-01-16

Reject